# Genome-Wide Identification and Expression Analysis of *XTH* Gene Family during Flower-Opening Stages in *Osmanthus fragrans*

**DOI:** 10.3390/plants11081015

**Published:** 2022-04-08

**Authors:** Yang Yang, Yunfeng Miao, Shiwei Zhong, Qiu Fang, Yiguang Wang, Bin Dong, Hongbo Zhao

**Affiliations:** 1Zhejiang Provincial Key Laboratory of Germplasm Innovation and Utilization for Garden Plants, Hangzhou 311300, China; 2019105051027@stu.zafu.edu.cn (Y.Y.); 2019105052024@stu.zafu.edu.cn (Y.M.); zsw1105@zafu.edu.cn (S.Z.); fangqiu@zafu.edu.cn (Q.F.); wangyiguang1990@zafu.edu.cn (Y.W.); 2Key Laboratory of National Forestry and Grassland Administration on Germplasm Innovation and Utilization for Southern Garden Plants, Hangzhou 311300, China; 3School of Landscape Architecture, Zhejiang Agriculture and Forestry University, Hangzhou 311300, China

**Keywords:** *Osmanthus fragrans*, xyloglucan endoglycolase/hydrolase, flower opening, ambient temperature

## Abstract

*Osmanthus fragrans* is an aromatic plant which is widely used in landscaping and garden greening in China. However, the process of flower opening is significantly affected by ambient temperature changes. Cell expansion in petals is the primary factor responsible for flower opening. Xyloglucan endoglycolase/hydrolase (XTH) is a cell-wall-loosening protein involved in cell expansion or cell-wall weakening. Through whole-genome analysis, 38 *OfXTH* genes were identified in *O. fragrans* which belong to the four main phylogenetic groups. The gene structure, chromosomal location, synteny relationship, and cis-acting elements prediction and expression patterns were analyzed on a genome-wide scale. The expression patterns showed that most *OfXTHs* were closely associated with the flower-opening period of *O. fragrans*. At the early flower-opening stage (S1 and S2), transcriptome and qRT-PCR analysis revealed the expression of *OfXTH24*, *27*, *32*, *35*, and *36* significantly increased under low ambient temperature (19 °C). It is speculated that the five genes might be involved in the regulation of flower opening by responding to ambient temperature changes. Our results provide solid foundation for the functional analysis of *OfXTH* genes and help to explore the mechanism of flower opening responding to ambient temperature in *O. fragrans*.

## 1. Introduction

Xyloglucan endotransglucosylase/hydrolase (XTH), a cell-wall-relaxation factor, modifies the cellulose–xylan complex structure through catalyzing the breaking and rejoining of xyloglucan molecules in the plant cell wall [1]. XTH belongs to the Glycoside Hydrolase Family 16 (GH16) of the Carbohydrate Active Enzyme Family Database (CAZy). XTH proteins contain a Glyco_hydro_16 protein domain and a XET_C domain [2,3]. Accumulating evidence has demonstrated that XTH proteins were related to the two kinks enzyme activity of xyloglucan transglycosylation (XET) and xyloglucan hydrolysis (XEH) [4]. In *Arabidopsis thaliana*, *XTH* gene family can be divided into four groups, and the different groups have different functions or enzyme activity. For instance, Group I/II and Group IIIB genes show XET activity, while Group IIIA genes show XEH activity [2,5].

Increasing reports showed that XTH family genes play important roles in many processes of plant growth and development. During root development stages, high expression of *SlXTH1* can be increased by XET activity in the hypocotyl elongation zone and affect the hypocotyl growth and cell wall extensibility in tomato [6]. Similarly, *AtXTH21* gene played a principal role in the growth of the primary roots by altering the deposition of cellulose and the elongation of the cell wall [7]. In addition, *AtXTH9* was positively correlated with cell elongation of the inflorescence stem [8]. In flowering plants, flower opening is dependent on petal growth, which is closely related to the cell expansion. In *Ipomoea nil*, *InXTH2*, *InXTH3*, and *InXTH4* were upregulated by dark period and their expression was positively correlated with the rate of flower opening [9]. *EgXTH1* from *Eustoma grandiflorum* was involved in cell expansion, and the expression was increased with the flower-opening process compared with the control [10]. In *Rosa hybrida*, *RhXTH1* were relevant to the typical growth in different part of petals, and also increased markedly during petal growth [11]. Conversely, if *XTH* genes are not expressed or inhibited, the petal growth will not continue, thereby effectively inhibiting flower opening [12]. Therefore, this evidence suggests that *XTH* genes are an important regulator, and significantly influence petal cell expansion during the flower-opening process of most species.

*Osmanthus fragrans* is one of the Chinese top ten traditional flowers, known for its strong fragrance, and is widely cultivated as a garden tree in most Asian countries due to its high ornamental value and aromatic scent. However, the changes in ambient temperature significantly affected the opening and longevity of the flower [13]. In our previous research, we found that the flower opening is significantly promoted by low ambient temperature 19 °C in *O. fragrans*, and the flower-opening process is accompanied by petal cells significant expansion [14,15]. To clarify whether *OfXTH* genes regulate flower opening in response to ambient temperature, the expression profiles of *OfXTH* genes were analyzed during flower-opening process under different temperature treatment. Our results could deepen understanding of the *OfXTH* gene family and help to explore the mechanism of *O. fragrans* petal expansion responding to ambient temperature changes.

## 2. Results

### 2.1. Identification of XTH Gene Family in O. fragrans

Through the HMM search with the XTH protein domains (PF00722 and PF06955) and BLASTp with 33 AtXTH proteins, a total of 38 *OfXTH* genes were identified from the *O. fragrans* genome. Then, the characteristics of OfXTH proteins were investigated, including protein length, molecular weight (MW), and theoretical isoelectric point (pI), by the online software Expasy. The signal peptide (SP) and subcellular localization were predicted by the online program SignalP 4.1 Service and Plant-mPLoc. The results showed that the length of amino acids was between 234 and 355, and the molecular weight of the proteins was between 26.8 and 40.4 kD (Table 1), corresponding with the protein length. The theoretical isoelectric point varied from 4.83 to 9.47; this is possibly due to the different polarities of the amino acids. In addition, a total of 29 OfXTH proteins were predicted to contain a signal peptide, while the other nine proteins had no signal peptide. Additionally, the subcellular localization prediction revealed that 32 OfXTH proteins were located in the cell wall, and six proteins were predicted to locate in the cell wall or cytoplasm (Table 1).

### 2.2. Phylogenetic Analysis of XTH Proteins

The phylogenetic tree was constructed using 38 OfXTHs from *O. fragrans*, 33 AtXTHs from *A. thaliana*, and 29 OsXTHs from *Oryza sativa* to analyze the evolutionary relationship (Figure 1). According to phylogenetic analysis, thirty-eight OfXTH proteins were classified into four groups in *O. fragrans*. Twenty-six OfXTHs belong to Group I/II. Group IIIA contained five OfXTH proteins. Group IIIB contained six OfXTH proteins. Only one protein, *OfXTH28*, belonged to Ancestral Group.

### 2.3. Gene Structure and Motif Analyses

Gene Structure Display Server (GSDS) was used to investigate the *OfXTHs* gene structure. The number of exons in thirty-seven *Of**XTHs* ranged widely within 3–5 (Figure 2). Only *OfXTH23* contained two exons. The conservative motif, ExDxE, was present in every OfXTH protein. Motifs 3, 5, 8, and 9 were highly conserved in all proteins. Group I/II genes contained common motifs 2–9, while OfXTH16 lacked motifs 4, 6, 7, and OfXTH34 lacked motif 7. In addition, motif 10 was only present in proteins of Group I/II. Motifs 1–5 and 7–9 were found in Group IIIA, and motifs 1–3, 5, 8 and 9 presented in Group IIIB. In Ancestral Group, OfXTH28 shared the same motif with other OfXTHs, except for motif 2.

### 2.4. Chromosomal Location and Synteny Analysis

As shown in Figure 3, 38 *OfXTH**s* were distributed on the 17 chromosomes. Notably, seven genes, *OfXTH14*–*20*, were densely located on the Chr08, which was the chromosome distributing with the largest number of *OfXTH* genes. No *OfXTH* gene located on Chr05, Chr07, Chr14, Chr17, and Chr21. In addition, the other chromosomes contained 1–4 *OfXTH* genes. The locations of *OfXTH* genes on the chromosomes are described in detail in the mentary Materials (Appendix A). Duplication events are related to plant evolutionary patterns. In this study, thirty segmentally duplicated genes were identified, and no tandem duplicated gene was found (Figure 4). All the synteny gene pairs were in the same cluster group. The substitution rates of Ka (non-synonymous) and Ks (synonymous) were calculated to estimate the selection pressure and determine the divergence time of the duplicated events. A total of 20 segmentally duplicated gene pairs were identified from 38 *OfXTHs* (Table 2). Interestingly, the Ka/Ks ratios of these 20 *OfXTHs* segmentally duplicated gene pairs were less than 1, suggesting that all segmentally duplicated gene pairs were purified selected instead of positively selected. The divergence time of *OfXTH* genes suggested that the duplicated events dated back to 59.93 million years ago (Mya) and lasted up to 0.31 Mya, and most of the duplicated events occurred between 5 and 35 Mya (Table 2).

### 2.5. Analysis of Cis-Acting Regulatory Elements in the Promoters of OfXTH Genes

For further understanding of the cis-acting regulatory elements in the promoter of *OfXTHs*, 2 kb upstream sequences of the transcription initiation site were subjected to PlantCARE. A large number of cis-acting elements in response to hormones were found in *OfXTH* promoter sequences (Figure 5), including ABA (abscisic acid), MeJA (methyl jasmonate), GA (gibberellin), SA (salicylic acid), and auxin-responsive elements. Except for *OfXTH15* and *OfXTH32*, at least one ABA-responsive element was found in other *OfXTH* promoter sequences. Cis-acting regulatory elements involved in MeJA were presented in 29 *OfXTHs*. In addition, other cis-acting regulatory elements, including anaerobic induction, drought inducibility, low temperature, etc., were found in 33, 26, and 11 *OfXTH* promoters, respectively.

### 2.6. OfXTH Genes Expression Analysis during Flower-Opening Stages

To understand which *OfXTH* genes are involved in the regulation of the flowering processes, the quantitative real-time PCR was performed to detect *OfXTH* expression patterns during different flowering stages in *O. fragrans*. Flower bud phenotype at different stages are shown in Appendix A. According to the expression profiles, *OfXTH**s* can be classified into three major groups (Figure 6). The first group, including four genes (*OfXTH8*, *12*, *26*, and *37*), exhibited that these *OfXTH* genes were highly expressed in the S0 period and then decreased at other flowering stages (Figure 6A). Twenty-eight genes belong to the second group, including *OfXTH**1*–*7*, *9*–*15*, *17*, *21*–*24*, *27*–*32*, *34*–*36*, and *38*. The expression pattern analysis showed that these *OfXTH* genes significantly upregulated in the S1 and S2 period, or only expressed in the S2 period (Figure 6B). Finally, six *OfXTH* genes were categorized into the third group, including *OfXTH**16*, *18*–*20*, *25*, and *33*. This group of genes have a high expression level in flowering periods S4 and S5 (Figure 6C).

### 2.7. Transcriptome Analysis of OfXTH Genes under Different Ambient Temperature Treatment

From our previous investigated results, ambient temperature of 19 °C significantly promoted the flower-opening process in *O. fragrans*, while ambient temperature of 23 °C inhibited the flowering process [15]. In our study, the flower buds were also treated under 19 °C and 23 °C conditions. In the 19 °C treatment for 2 and 5 days, the traits of flower buds were reached at S1 and S2 stage, respectively. However, the flower buds still remained in the S1 stage when treated at 23 °C for either 2 days or 5 days. Therefore, the flower buds were collected at the second and fifth day under 19 °C and 23 °C treatment for transcriptome sequencing, and then we explored which *OfXTHs* are involved in the regulation of flower-opening processes responding to ambient temperature changes. Based on log2(ratio of abundance) ≤−1 or ≥1 with FDR < 0.001, a total of 5702 and 8082 differentially expressed genes (DEGs) were identified in 2 d, 19 °C vs. 2 d, 23 °C and 5 d, 19 °C vs. 5 d, 23 °C, respectively (Appendix A). Among these obtained DEGs, five and sixteen differentially expressed *OfXTH* genes were identified at the second and fifth day under 19 °C treatment relative to the control 23 °C, respectively (Table 3). The differentially expressed *OfXTH* genes including *OfXTH24*, *27*, *32*, *35*, and *36* were all induced by 19 °C at the second day. Compared with the control of 23 °C at the fifth day, fifteen upregulated *OfXTH* genes (*OfXTH4*–*7*, *12*, *14*, *23*–*25*, *27*, *28*, *31*–*33*, *35*, and *36*) were identified under 19 °C treatment. The expression of five *OfXTH* genes, including *OfXTH24*, *27*, *32*, *35*, and *36,* were significantly increased at the second and fifth day. Then, qRT-PCR was employed to validate the differentially expressed *OfXTH* genes. Consistent with the transcriptome profile, the expression of *OfXTH24*, *27*, *32*, *35*, and *36* were strongly induced by 19 °C (Figure 7).

## 3. Discussion

### 3.1. The Characteristics of XTH Gene Family in O. fragrans

In this study, a total of 38 *OfXTH* genes were identified, and the corresponding proteins were analyzed for their physical–chemical properties in *O. fragrans* genome (Table 1). All of the OfXTH proteins were predicted to be located in the cell wall, which is consistent with the function where XTH proteins are involved in cell-wall reconstruction [5,16]. A signal peptide was shown in most OfXTH proteins. When the signaling peptide of *D**kXTH6* was absent, the subcellular localization of *D**kXTH6* protein was transformed from the cell wall to the whole cell [17]. It implied that a signal peptide played an important role in the transmembrane transport of XTH protein and guided the protein to the plant cell wall. All of the *OfXTH* genes were categorized into four subfamilies by polygenetic analysis (Figure 1). The number of 26, 5, and 6 XTHs were contained in Group I/II, IIIA, and IIIB, respectively, whereas only one XTH was contained in Ancestry Group. Compared with *A**. thaliana*, the numbers of XTH proteins in Group I/II, Group IIIA, and Group IIIB were increased, while Ancestral Group was decreased in *O. fragrans* (Figure 1). This might be due to the gene duplication or loss during the evolution of the *XTH* gene family [18]. Most *OfXTHs* are involved in multiple segmental duplications, which might contribute to the expansion and evolution of the OfXTH gene family [19]. It might be the reason why the number of the *XTH* family members in *O. fragrans* exceeds other species, such as *A. thaliana* (33) [20], *O. sativa* (29) [21], and *Hordeum vulgare* (24) [22]. The Ka/Ks values of all duplicated *OfXTH* gene pairs were <1.00 (Table 2), suggesting that the evolution of duplicated *OfXTH* gene pairs occurred through purifying selection and evolved slowly in *O. fragrans*.

Cis-acting elements are specific DNA sequences connected in tandem with structural genes and are binding sites for transcription factors [23]. Through analysis of *OfXTH* promoters, we found that ABA and MeJA responsiveness elements presented in most *OfXTH* promoters (Figure 5), which suggested that *OfXTH* genes might be regulated by abscisic acid and jasmonic acid, similar to the *XTH* genes in sweet cherry [24]. Recent research showed that the expression of *PavXTH15* was elevated by the phytohormone ABA and MeJA in sweet cherry [24]. Cis-acting elements involved in SA responsiveness were found in *OfXTH* promoters, indicating that the *OfXTH* gene might be the target gene of the SA signaling pathway. Similar findings reported previously showed that the accumulation of SA suppressed cell division and elongation through reducing *AtXTH8* and *AtXTH31* expression [25]. Besides responding to plant hormones, the promoters of *OfXTHs* also contain cis-acting regulatory elements involved in drought inducibility, low-temperature responsiveness, defense, and stress responsiveness. The result suggested that *OfXTH* may respond to environmental factors and be related to biotic and abiotic stress resistance. Similar results have been reported before. Overexpression of *DkXTH1* enhances tolerance to abiotic stress salt and drought stresses in transgenic *Arabidopsis* plants with respect to root and leaf growth, and survival [26]. The *XTH19* mutant showed reduced freezing tolerance after both cold and sub-zero acclimation [27].

### 3.2. OfXTHs Are Closely Associated with the Flower-Opening Period of O. fragrans

The flower-opening process is accompanied by the rapid expansion of petal cells. The size of the adaxial petal epidermal cells and the abaxial petal epidermal cells increased by 37.5% and 6.5% from S1 to S2, respectively [15]. Previous works have reported that XTHs participated in flower-opening processes in many species. *DcXTH2* and *DcXTH3* transcripts were markedly accumulated in *Dianthus caryophyllus* petals of opening flowers and showed high XET activity in petal claw, which is the main part of carnation petal elongation [28]. *LhXTH1* transcript levels in the petals markedly increased during *Lilium* flower opening and were higher in adaxial epidermal cells contributing to petal expansion [29]. In addition, it is reported that *XTH* plays an important role in the rapid petal growth period of *E. grandiflorum* [10]. In our study, we focused on the *OfXTH* expression changes in the S1 and S2 periods. A total of 28 *OfXTH* genes were found and significantly upregulated in S1 and S2 periods relative to the S0 period (Figure 6B), implying that these genes might be involved in petal cell expansion and contribute to the regulation of flower opening in *O. fragrans*. In addition, it has been reported that the expression of *RbXTH1* and *RbXTH2* in *Rosa bourboniana* leads to petal abscission [30]. Overexpression of persimmon *DkXTH8* causes cells to be easily destroyed and accelerates leaf senescence in transgenic plants [31]. Six *OfXTH* genes have high expression level in the flowering periods of S4 and S5 (Figure 6C); we hypothesized that these *OfXTH* genes may contribute to the cellular senescence.

### 3.3. OfXTHs Response to Ambient Temperature in Regulation of Flower Opening

Flowering in many species is influenced by ambient temperature. In *Arabidopsis*, the transition to flowering is delayed in low ambient temperatures [32]. Phased constant temperature treatment and a lower temperature could shorten the flower differentiation process and promote the flowering of *Crocus sativus* [33]. Similarly, the change in ambient temperature affected flower-opening processes significantly in *O. fragrans*. The flower opening is significantly promoted by low ambient temperature (19 °C) in *O. fragrans*, and the flower buds could not open until ambient temperature was below 23 °C [15]. However, there are fewer reports on the role of *XTHs* in responding to low ambient temperature; most studies focused on the high temperature regulating cell growth. For example, the expression of *AtXTH9* and *AtXTH11* was strongly increased by high temperature in stem and root, respectively, to accelerate cell elongation [8,34]. Through transcriptome analysis, we found that five and sixteen *OfXTH* genes were differentially expressed at the second and fifth day under 19 °C treatment compared with the control 23 °C, respectively (Table 3). Specifically, *OfXTH24*, *27*, *32*, *35*, and *36* were significantly upregulated at both the second and fifth day after 19 °C treatment, which indicated that *OfXTH24*, *27*, *32*, *35*, and *36* might respond to ambient temperature changes and contribute to the process of low ambient temperature accelerated flower opening of *O. fragrans.*

## 4. Conclusions

In this study, thirty-eight *OfXTH* genes were identified from the *O. fragrans* genome database. All of the OfXTH proteins were predicted to locate in the cell wall and were divided into four groups. Multiple segment duplication happened in the evolution of the *OfXTH* genes and contributed to the expansion of *OfXTH* gene family. The expression profiles showed that most *OfXTHs* are closely associated with the flower-opening processes in *O. fragrans*. Using transcriptome and qRT-PCR analysis, five *OfXTH* genes (*OfXTH24*, *27*, *32*, *35*, and *36*) were identified and significantly upregulated at the S1 and S2 stages under 19 °C treatment, indicating that these *OfXTH* genes might be key regulators responding to ambient temperature to modulate petal cell expansion during flower opening. In conclusion, the present research increased our knowledge of the role of the XTH gene family in flower-opening processes in *O. fragrans*.

## 5. Materials and Methods

### 5.1. Identification of XTH Gene Family in O. fragrans

The whole genome of *O. fragrans* was obtained from the website http://117.78.20.255/ [35] (accessed on 17 March 2022). The protein sequences of *AtXTH* (*A. thaliana*) and *OsXTH* (*O**. sativa*) were downloaded from the Ensembl Plants database (http://plants.ensembl.org (accessed on 17 March 2022)). Two programs of TBtools1.6 [36], Simple HMM Search and BLASTp, were used to identify *OfXTH* family genes. To be specific, the Simple HMM Search program was used to search the two XTH protein domains, PF00722 (ID: Glyco_hydro_16) and PF06955 (ID: XET_C), from the database [37]. Thirty-three AtXTH proteins were used as queries to blast the *O. fragrans* genome database and identify XTH homologs in *O. fragrans*, with the parameters E-value < 10^−15^ and identity > 50%. Only protein sequences present in both outcomes were regarded as OfXTH proteins and used for further analysis. SMART (http://smart.embl-heidelberg.de/ (accessed on 17 March 2022)) was used to check the conserved XTH domains Glyco_hydro_16 and XET_C with default parameters. The ProtParam tool of online software Expasy (http://web.expasy.org/protparam/ (accessed on 17 March 2022)) was used to determine the physicochemical parameters of OfXTHs protein sequence. SignalP 4.1 Service (http://www.cbs.dtu.dk/services/SignalP-4.1/ (accessed on 17 March 2022)) was used to predict the signal peptide. Plant-mPLoc (http://www.csbio.sjtu.edu.cn/bioinf/plant-multi/ (accessed on 17 March 2022)) was used to predict the sub-cellular localizations.

### 5.2. Phylogenetic Analysis of XTH Proteins

The Clutsalx-v1.83 program was used to align the XTH sequences from *O. fragrans*, *A. thaliana*, and *O. sativa*. MEGA6.0 was then used for phylogenetic analysis via the neighbor-joining (NJ) method with 1000 bootstrap replicates. The online software ITOL (https://itol.embl.de/ (accessed on 17 March 2022)) was used to annotate the phylogenetic tree.

### 5.3. Motif Identification and Gene Structure Analysis

MEME Suite 5.3.0 (http://meme-suite.org/tools/meme (accessed on 17 March 2022)) was used to identify the motifs of OfXTH proteins, at a maximum motif number of 10, with a minimum and maximum width of six and 50, respectively. The online software GSDS (http://gsds.gao-lab.org/ (accessed on 17 March 2022)) was used to predict the gene structure of *OfXTHs*.

### 5.4. Chromosomal Location and Synteny Analysis

MapChart (version 2.32) was used to predict the distribution of OfXTH genes on chromosomes. Quick Run MCScanX Wrapper program in TBtools1.6 was used to assess gene duplication events of *OfXTH**s*. The synonymous (Ka) and non-synonymous (Ks) substitution rate were calculated by the Simple Ka/Ks Calculator program of TBtools1.6. The divergence was calculated as follows: T = Ks/2λ, where λ = 1.5 × 10^−8^ s for dicots [38]. The selection mode was identified using Ka/Ks value (Ka/Ks ratio >1, <1, and =1 represented positive selection, negative selection, and neutral selection, respectively) [38].

### 5.5. Analysis of Cis-Acting Regulatory Elements in Promoter

The upstream sequences (2 kb) of the OfXTH-coding sequences were obtained from the genome database of *O. fragrans* [35]. PlantCARE (http://bioinformatics.psb.ugent.be/webtools/plantcare/html/ (accessed on 17 March 2022)) was used to identify cis-acting regulatory elements in *OfXTH* genes promoter.

### 5.6. Plant Materials and Ambient Temperature Treatment

*O. fragrans* plants, cultivar “Yanhonggui”, were potted and grown in the Zhejiang Agriculture and Forestry University Osmanthus Resource Nursery. The flower buds at different stages were collected between 9.00–10.00 a.m. every day, including S0 (the outer bud scales unfurled and the inner bud scales still furled); S1 (the bud became globular-shaped and the inside bracts covering the inflorescence was visible); S2 (the inflorescence burst through bracts and the florets closely crowded); S3 (the florets are bud-shaped and the pedicels elongate); S4 (initial flowering stage); S5 (full flowering stage). The flower bud phenotype of *O. fragrans* at different stages are shown in Appendix A. In addition, the plants in S0 stage with similar size were selected and treated with different temperatures (19 °C and 23 °C) in a growth chamber. When treated at 19 °C for 2 days and 5 days, the flower buds reached S1 and S2 stages, respectively, while flower buds still remained in S1 stage when treated at 23 °C for 2 days and 5 days. The phenotype of flower buds in 19 °C and 23 °C are shown in Appendix A. The relative humidity was about 70% and the photoperiod was a 12 h light/12 h dark regime with illumination of 10,000 lux. Each treatment included three independent biological replications. Samples of each stage were collected and stored at −80 °C before total RNA extraction.

### 5.7. Total RNA Extraction and Transcriptome Sequencing

RNA was isolated from 300 mg of flower buds at S1 and S2 period under 19 °C and 23 °C treatment using the EasyPure Plant RNA Kit (Tiangen, Beijing, China). The quality of total RNA satisfied the standards 1.8 ≤ OD_260/280_ ≤ 2.2 and OD_260/230_ ≥ 1.8. Five μg of total RNA was employed for cDNA libraries preparation and RNA deep sequencing (6 G × 250 bp) by Novogene Biological Information Technology Company (Beijing, PR China). Three biological replications were performed for the transcriptome analysis. The FPKM (fragments per kilo base of exon per million fragments mapped) was used to estimate gene expression levels and identify differentially expressed genes among 2 d, 19 °C vs. 2 d, 23 °C, and 5 d, 19 °C vs. 5 d, 23 °C. The differentially expressed genes were analyzed according to the log2(ratio of abundance) threshold, the value ≥1.0 was defined as upregulation, and ≤−1.0 was defined as downregulation, along with FDR <0.001.

### 5.8. Real-Time Quantitative PCR

PrimeScript RT Reagent Kit (Takara, Dalian, China) was used for first-strand cDNA synthesis following the manufacturer’s instructions. LightCycler480II Real-Time PCR System was used for qRT-PCR analysis. The reaction mixture (10 μL total volume) consisted of 5 μL SYBR Premix Ex Taq (Takara, Dalian, China), 20 ng cDNA template, forward and reverse primers (10 microns/L) of 0.4 μL, was and filled with ddH_2_O to 10 μL. Three biological replications were performed for each treatment. The relative expression of the target gene was calculated using the 2^−^^ΔΔCT^ method. OfACT was used as a reference gene [39]. Primers of *OfXTH* genes used in RT-qPCR are shown in Appendix A.

## Figures and Tables

**Figure 1 plants-11-01015-f001:**
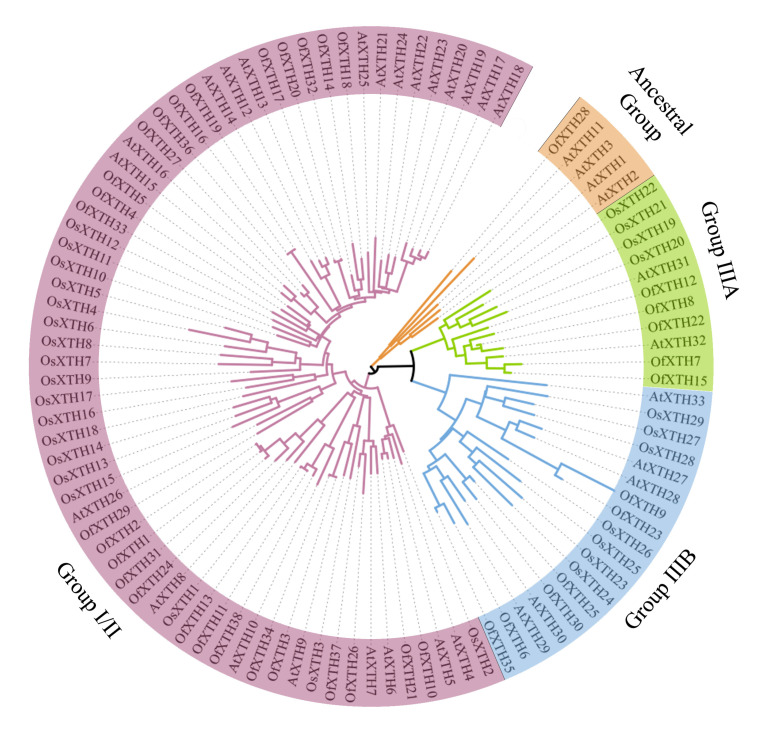
Phylogenetic analysis of XTH proteins from *O. fragrans*, *A. thaliana,* and *O. sativa*. Each subgroup is distinguished by a different color.

**Figure 2 plants-11-01015-f002:**
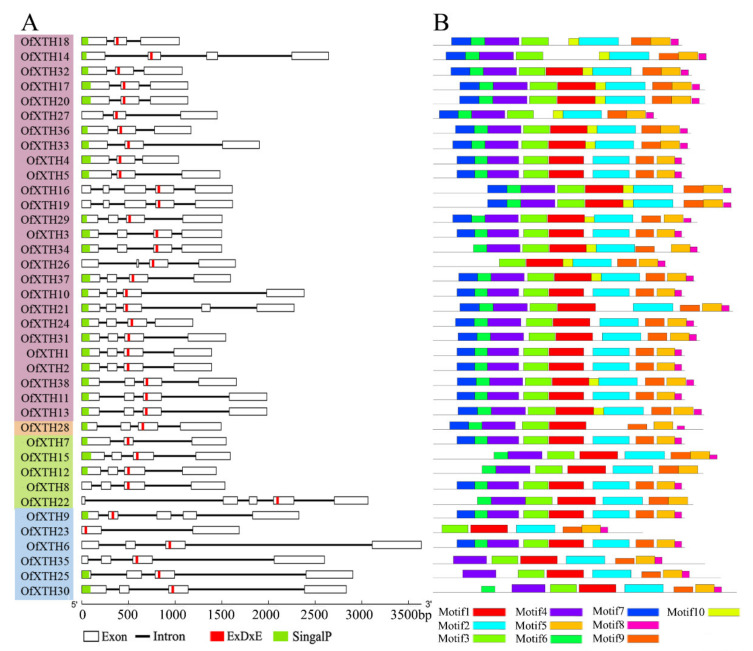
Gene structure and motif distribution of OfXTH proteins. (**A**) Gene structure, putative signal peptide sequences, and conservative motif sequence. Each subgroup is distinguished by a different color. White boxes and black lines represent exons and introns, respectively. The green and red strips indicate putative signal peptide sequences and conservative motif sequence, respectively. (**B**) The motif distribution. Motifs 1–10 in the OfXTH proteins are represented by different colored boxes.

**Figure 3 plants-11-01015-f003:**
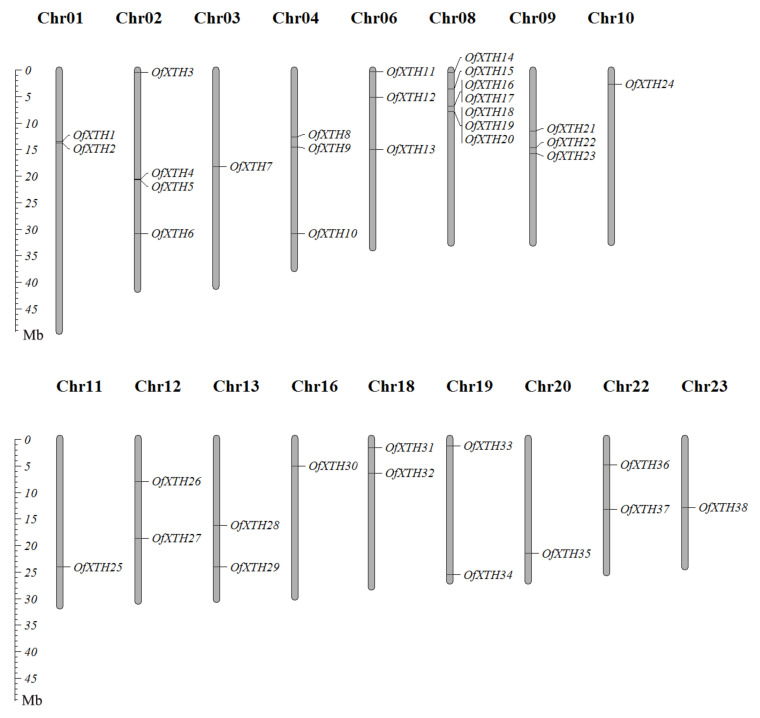
Chromosomal distribution of *OfXTH* genes.

**Figure 4 plants-11-01015-f004:**
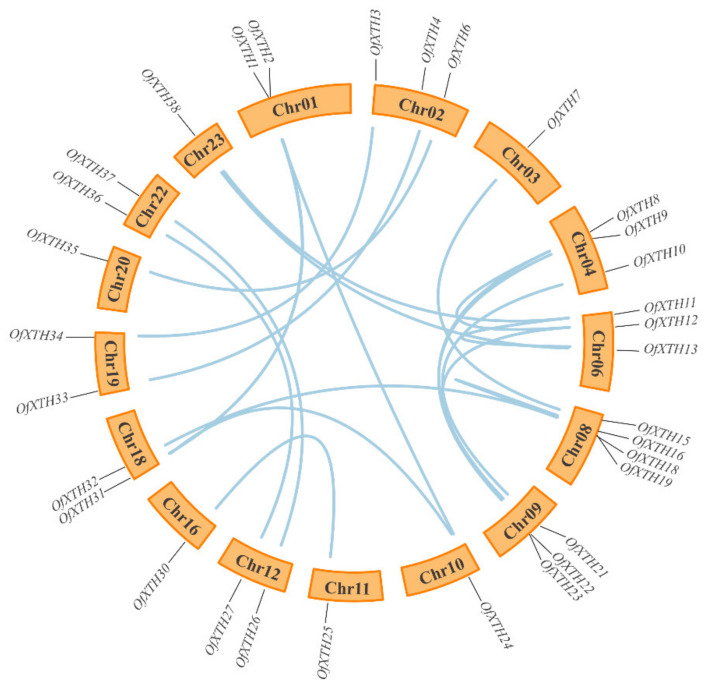
Synteny analysis of 38 *OfXTH* genes in *O. fragrans*. Orange rectangles represent chromosomes of *O. fragrans*, and blue lines represent segment duplicate gene pairs.

**Figure 5 plants-11-01015-f005:**
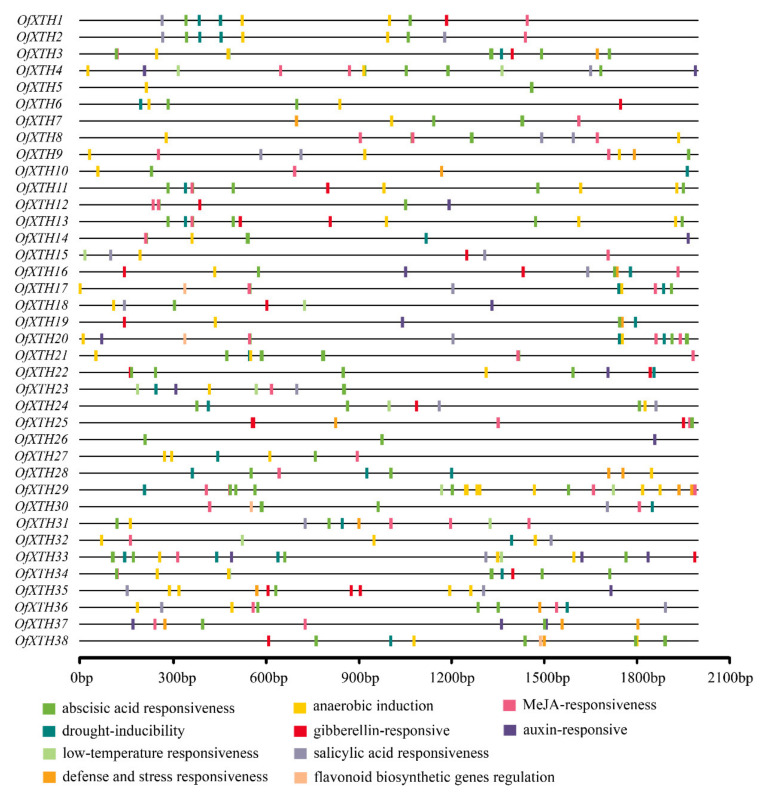
Prediction of cis-responsive elements in the *OfXTH* promoters. Different cis-responsive elements are represented by different colored boxes.

**Figure 6 plants-11-01015-f006:**
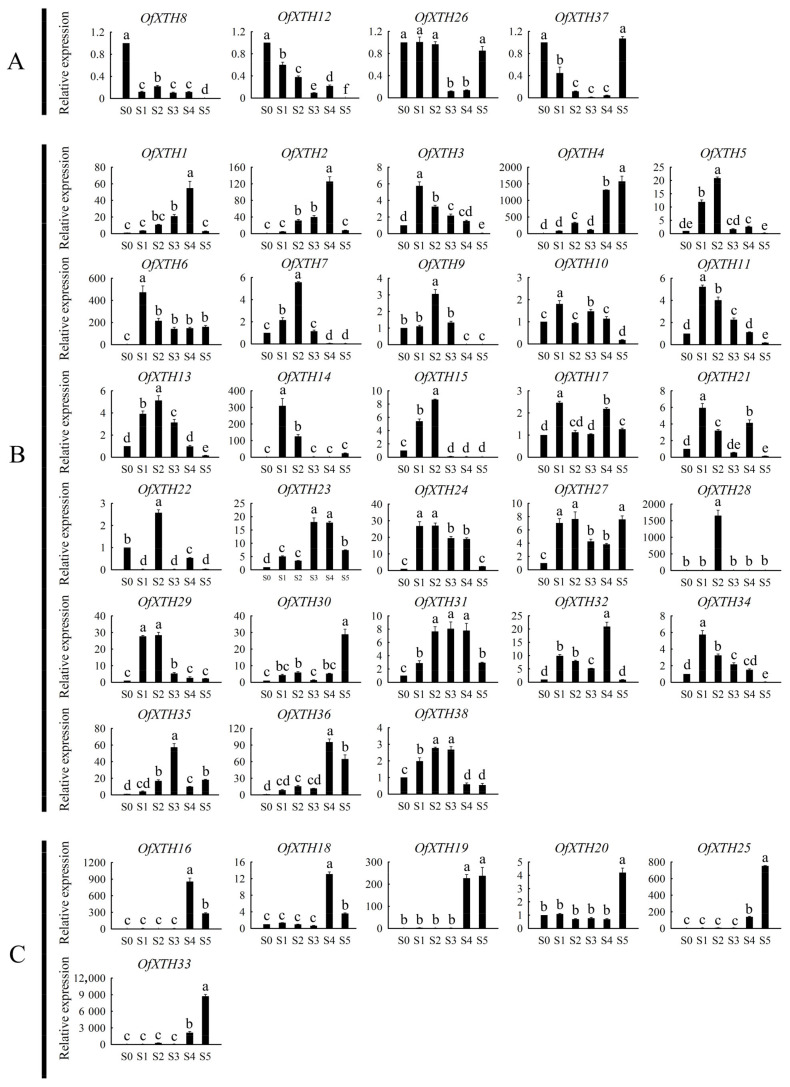
Expression profiles of *OfXTHs* at different flowering stages. Significant differences are identified by SPSS (version 22) with Duncan’s test (*p* < 0.05) and are represented by different letters above the error bars. (**A**) *OfXTH* genes highly expressed in the S0 period. (**B**) *OfXTH* genes upregulated in the S1 and S2 period. (**C**) *OfXTH* genes upregulated in the S4 and S5 period.

**Figure 7 plants-11-01015-f007:**
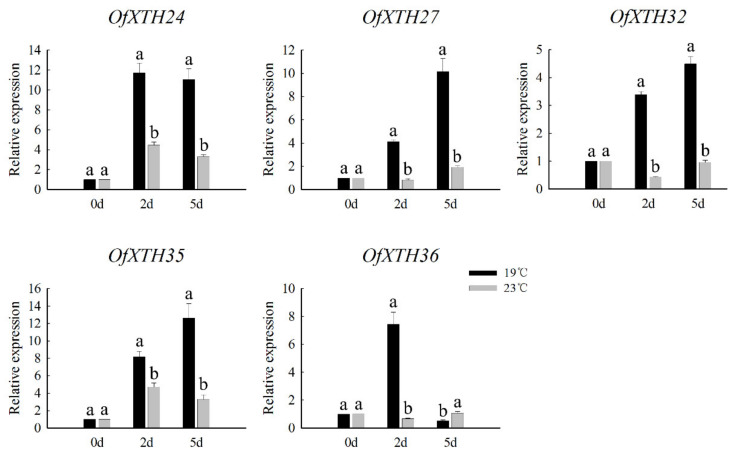
The qRT-PCR analysis of five differentially expressed *OfXTH* genes treated at 19 °C and 23 °C. Significant differences are identified by SPSS (version 22) with Duncan’s test (*p* < 0.05) and are represented by different letters above the error bars.

**Table 1 plants-11-01015-t001:** The characteristic of *OfXTH* genes in *O. fragrans*.

Gene	Length	MW	PI	SP	Subcellular Localization
*OfXTH1*	298	34,863.67	4.83	28	cell wall
*OfXTH2*	298	34,958.87	5.03	28	cell wall
*OfXTH3*	298	33,595.75	5.57	29	cell wall
*OfXTH4*	289	32,550.53	8.85	32	cell wall
*OfXTH5*	305	34,591.01	6.51	26	cell wall
*OfXTH6*	338	39,335.44	9.05	—	cell wall
*OfXTH7*	255	29,221.92	9.43	20	cell wall
*OfXTH8*	260	29,772.14	8.24	—	cell wall
*OfXTH9*	355	40,429.32	8.84	20	cell wall
*OfXTH10*	294	34,293.78	8.08	23	cell wall
*OfXTH11*	294	34,213.84	8.84	25	cell wall/cytoplasm
*OfXTH12*	293	33,385.14	5.74	21	cell wall
*OfXTH13*	294	34,263.86	8.83	25	cell wall
*OfXTH14*	297	33,314.48	5.21	17	cell wall
*OfXTH15*	308	35,606.44	9.47	33	cell wall
*OfXTH16*	324	37,516.58	9.03	—	cell wall
*OfXTH17*	295	33,485.54	6.89	32	cell wall/cytoplasm
*OfXTH18*	270	30,312.99	5.23	23	cell wall
*OfXTH19*	324	37,516.58	9.03	—	cell wall
*OfXTH20*	295	33,463.49	6.43	32	cell wall/cytoplasm
*OfXTH21*	325	37,273.13	8.93	24	cell wall/cytoplasm
*OfXTH22*	282	31,918.7	8.69	—	cell wall
*OfXTH23*	234	26,856.03	8.79	—	cell wall
*OfXTH24*	295	34,651.93	5.14	26	cell wall
*OfXTH25*	321	37,347.8	8.96	26	cell wall
*OfXTH26*	261	30,242.91	5.98	—	cell wall
*OfXTH27*	250	28,403.93	9.1	—	cell wall
*OfXTH28*	302	35,166.08	5.7	20	cell wall
*OfXTH29*	295	33,937.19	6.75	18	cell wall
*OfXTH30*	339	38,895.06	6.4	31	cell wall
*OfXTH31*	298	34,746.73	5.02	28	cell wall
*OfXTH32*	289	32,124.53	5.1	23	cell wall/cytoplasm
*OfXTH33*	287	32,580.54	8.78	25	cell wall
*OfXTH34*	298	33,611.81	5.57	29	cell wall
*OfXTH35*	304	34,732.12	9.08	—	cell wall
*OfXTH36*	288	32,847.03	9.22	21	cell wall/cytoplasm
*OfXTH37*	293	33,556.83	6.31	30	cell wall
*OfXTH38*	294	34,368.01	8.69	25	cell wall

**Table 2 plants-11-01015-t002:** Ka/Ks analysis and estimated divergence time of *OfXTHs*.

Duplicated Gene Pairs	Ka	Ks	Ka/Ks	Type of Selection	Divergence Time (Mya)
*OfXTH1-OfXTH24*	0.118	0.987	0.119	Purifying	32.91
*OfXTH1-OfXTH31*	0.017	0.346	0.050	Purifying	11.55
*OfXTH2-OfXTH31*	0.023	0.355	0.065	Purifying	11.82
*OfXTH4-OfXTH33*	0.221	1.798	0.123	Purifying	59.93
*OfXTH6-OfXTH35*	0.083	0.394	0.212	Purifying	13.14
*OfXTH7-OfXTH15*	0.059	0.371	0.158	Purifying	12.36
*OfXTH8-OfXTH12*	0.121	0.666	0.181	Purifying	22.21
*OfXTH8-OfXTH22*	0.051	0.264	0.194	Purifying	8.82
*OfXTH9-OfXTH23*	0.262	0.568	0.461	Purifying	18.93
*OfXTH10-OfXTH21*	0.027	0.274	0.100	Purifying	9.13
*OfXTH11-OfXTH13*	0.003	0.011	0.267	Purifying	0.36
*OfXTH11-OfXTH38*	0.034	0.205	0.166	Purifying	6.84
*OfXTH12-OfXTH22*	0.130	0.806	0.161	Purifying	26.87
*OfXTH13-OfXTH38*	0.038	0.194	0.194	Purifying	6.48
*OfXTH16-OfXTH19*	0.000	0.009	0.000	Purifying	0.31
*OfXTH18-OfXTH32*	0.143	0.958	0.150	Purifying	31.92
*OfXTH24-OfXTH31*	0.123	1.094	0.112	Purifying	36.48
*OfXTH25-OfXTH30*	0.078	0.345	0.226	Purifying	11.49
*OfXTH26-OfXTH37*	0.171	0.526	0.326	Purifying	17.54
*OfXTH27-OfXTH36*	0.038	0.222	0.169	Purifying	7.40

**Table 3 plants-11-01015-t003:** DEGs of *OfXTHs* treated at 19 °C and 23 °C for 2 and 5 days.

Gene	2 d, 19 °C FPKM	2 d, 23 °C FPKM	5 d, 19 °C FPKM	5 d, 23 °C FPKM	Log_2_ (Ratio of Abundance)			
					2 d, 19 °C vs. 2 d, 23 °C	FDR	5 d, 19 °C vs. 5 d, 23 °C	FDR
*OfXTH4*	3.97	0.38	18.99	0.05	3.37	9.4 × 10^−2^	8.69	1.6 × 10^−15^
*OfXTH5*	1.36	2.00	15.91	1.92	−0.56	2.8 × 10^−1^	3.05	5.9 × 10^−21^
*OfXTH6*	0.02	0.05	2.75	0.10	−1.64	1.0 × 10^0^	4.79	2.5 × 10^−8^
*OfXTH7*	1.90	1.47	3.10	0.18	0.38	3.8 × 10^−5^	4.09	4.9 × 10^−8^
*OfXTH12*	1.75	3.00	0.64	4.07	−0.78	1.5 × 10^−2^	−2.68	7.8 × 10^−7^
*OfXTH14*	1.98	0.85	88.96	1.97	1.22	7.9 × 10^−1^	5.50	3.6 × 10^−16^
*OfXTH23*	5.13	7.06	9.33	3.72	−0.46	2.4 × 10^−1^	1.33	7.6 × 10^−6^
*OfXTH24*	14.69	6.25	38.07	5.38	1.23	3.8 × 10^−9^	2.82	2.4 × 10^−58^
*OfXTH25*	3.22	3.90	19.84	3.93	−0.28	9.6 × 10^−2^	2.33	2.0 × 10^−26^
*OfXTH27*	58.21	6.08	87.85	5.47	3.26	3.1 × 10^−34^	4.01	3.1 × 10^−109^
*OfXTH28*	0.39	0.43	104.94	0.54	−0.15	8.0 × 10^−1^	7.60	4.0 × 10^−47^
*OfXTH31*	0.29	0.30	1.53	0.13	−0.06	8.9 × 10^−1^	3.55	1.7 × 10^−4^
*OfXTH32*	12.50	3.46	10.53	0.66	1.85	2.1 × 10^−30^	3.99	1.8 × 10^−4^
*OfXTH33*	0.31	0.04	1.71	0.02	2.83	4.5 × 10^−1^	6.21	1.8 × 10^−5^
*OfXTH35*	1.97	0.94	3.23	0.46	1.06	4.7 × 10^−4^	2.81	4.2 × 10^−7^
*OfXTH36*	23.03	2.76	70.54	2.32	3.06	3.1 × 10^−19^	4.93	2.3 × 10^−134^

## Data Availability

Data are contained within the article.

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
