# Peer review of "Genome-Wide Identification and Expression Analysis of XTH Gene Family during Flower-Opening Stages in Osmanthus fragrans"

_plants, 2022, doi:10.3390/plants11081015_

Round 1
Reviewer 1 Report
- The quality of all figures needs to be greatly improved.
- The authors need to indicate how many biological replications they performed for the transcriptome analysis. If multiple biological replications were performed, are the FPKM in Table 3 the mean of the biological replications?
- The method used for identification of DEGs between stages and treatments need to be included in the method.
- How long of the treatments with 19℃ and 23℃ should be involved in the method.
- In the legend of Figure S2, the authors mentioned “the buds treated at 23°C still stayed at S1 period all the time”, which suggested that the buds treated with 23°C cannot develop to S2 period. However, both of the table 3 and figure 7 indicated the RNAseq data and qRT-PCR were generated from buds at S2 period with the treatment of 23°C, how can the author collect the buds at S2 period under the treatment of 23°C, if the buds treated with 23°C cannot develop to S2 period?
Reviewer 2 Report
Abstract
"Through transcriptome and qRT-PCR analysis, OfXTH24, 27, 32, 35, and 36 were strongly upregulated by low ambient temperature (19°C) at S1 and S2 period and might involve in the regulation of flower opening by responding to ambient temperature changes." - this does not make sense, please rephrase the sentence.
Results
Please make clear whether the OfXTH were identified solely based on conserved domains (by the software TBtools1.6 and SMART) or BLAST search with known XTH genes.
Figure 1, if authors wanted to highlight the ancestral group then it is better to use that as root and redraw the tree.
Figure 3 and Figure 4 are redundant, The information provided in figure 3 is already there in figure 4. Therefore, I suggest removing Figure 3.
Prediction of cis-responsive elements is unreliable due to the very small size of motifs that could find in any sequence by chance.
Authors need to well elaborate the Table titles and figure legends. Please see Table 3 - DEGs of OfXTHs at S1 and S2 period under ambient temperature treatment. Here, explain all abbreviations.
Upregulation of OfXTH24, 27, 32, 35, and 36 by low ambient temperature make them candidates for further studies. The authors should confirm their expression in a flower petal and subcellular localization of the translated protein.
Reviewer 3 Report
The manuscript, plants-1664760, "Genome-wide identification and expression analysis of XTH gene family during flower opening stages in Osmanthus fragrans" by Yang et al is an interesting read on the gene expression analysis during flower opening in O. fragrans. The data are original and novel and increase knowledge of the flowering processes.
More specific comments are included below:
ABSTRACT
Line 15. which is widely used in landscaping
Line 19-20. Through whole-genome analysis, 38 OfXTH genes were identified in O. fragrans which belong to the four main phylogenetic groups.
Line 23-26. Our transcriptome and qRT-PCR analysis revealed the upregulation of OfXTH24, 27, 32, 35, and 36 in low ambient temperature (19°C) at the early flower opening stages (S1 and S2 stages). These genes might be involved in the regulation of flower opening by responding to ambient temperature changes.
INTRODUCTION
Line 64- 65. However, the changes in ambient temperature significantly affected the opening and longevity of the flower (please provide a few references here, preferably a review paper).
Line 68-71. Not clear; please rewrite this sentence.
RESULTS
Line 80. molecular weight (MW) and theoretical isoelectric point (pI)
Line 113-114. Not clear; please rewrite this sentence. [In Group IIIA proteins, Motif 1–5, and 7–9 were found in their protein sequence, while motifs (????) were no (not) regular (did you mean common??) in Group IIIB proteins].
Figure 6A. In the third graph, from the left (OfXTH26) Duncan's number doesn't look correct. For example, S1 is "ab" but looks taller than S2, which is "a" ("a" must be taller than "ab"). Or in the last graph (OfXTH37), we have "a" and "c" but we don't see the letter "b" (It should be in order).
Please check and make sure all your statistics, and correct the error bar in the figures.
Line 191. Remove "What's more"
Figure 7. Why "S0" does not have an error bar and Duncan's letter in all graphs?
DISCUSSION
The first paragraph in the discussion is pure results and it is a repeat of the result section. Please rewrite this paragraph (try to highlight your findings and compare them to other research, not just describe your results again).
Line 215. Remove "What’s more”
Line 216. These results showed that …
Line 220. Remove “usually” (it is a bit informal)
Line 225. OfXTH gene family (please add reference/s)
Line 231. “thus promoted fruit softening, which was (is) consistent with our findings”. I do not understand this conclusion. Could you please explain how “fruit softening” is related to “flower opening”? Please remove if not relevant to your results.
Line 235. and jasmonic acid in O. fragrans (please add reference/s)
Line 239-241. The promoters of OfXTHs contain cis-acting regulatory elements that are involved in anaerobic induction, drought inducibility, low-temperature responsiveness, defense and stress responsiveness, flavonoid biosynthetic genes regulation (please add reference/s).
Line 246-246. Please remove “O. fragrans is an aromatic tree and is widely applied in landscaping and garden greening in Asian areas”. This is more introductory
Line 264. that these OfXTH genes may contribute to the cellular
CONCLUSIONS
Line 289-291. Please remove the detailed results from your conclusions. In the conclusion section, the study's most important findings need to be highlighted (no need for the details).
Line 292-294. In conclusion, the present research increased our knowledge of the role of XTH gene family in flower opening processes in O. fragrans.
MATERIAL AND METHODS
Line 298. This link (http://117.78.20.255/profile-hic) did not open to the O. fragrans genome. Please check and update the link.
Line 307. Remove “An online website,”
Line 339. O. fragrans plants, cultivar Yanhonggui, were
Line 341. were collected at 9.00-10.00 (am or pm???; please add) every day
Line 346-348. The plants of similar size were selected and treated with different temperatures (19°C and 23°C) in a growth chamber.
Line 350. Each treatment included three independent biological replications.
Line 350-351. Remove “Before treatment, samples were recorded as S0”. You have already introduced S0 in Line 341, you don’t need to repeat it here again.
Line 351. 300 mg instead of 0.3 g
Line 356-357. Remove “ For transcriptome sequencing, 0.3 g flower buds at S1 and S2 stage were used to extract total RNA”. You have already mentioned this in line 351
Lin 357-359. Five μg of total RNA was used for cDNA libraries preparation and RNA deep sequencing (How deep? Please add more details; for example 100M × 150 bp) by Novogene Biological Information Technology Company (Beijing, PR China).
Line 363. “downregulation, along with P-value <0.001”. Why did not use the false discovery rate (FDR) instead of the P-value? For the RNA-seq data, FDR gives us more confidence than P-value. Could you please explain why you have ignored FDR?
Line 368. “2 μL cDNA template”. How much in “ng”? Please add the amount of RNA in ng instead of μL. For example “5 ng cDNA”.
Line 368-369. Replace “upstream and downstream primers” with “forward and reverse primers”
Line 371. “using the 2-â–³CT method”. It should be 2-△△CT . In your calculation, you should use this formula instead of 2-â–³CT.
Here are the details of 2-△△CT method:
ΔCT = CT target – CT reference
ΔΔCT = ΔCT treatment – ΔCT control
Please make sure that your qPCR calculations are correct.
Line 372. “Primers of OfXTH genes used in RT-qPCR were shown in Table S4 (?? Should be S3)”. I found the primer sequences in Table S3. Please check and make sure your citations to the supplementary data are correct.
I have checked the first three primers presented in Table S3 using Primer-BLAST in NCBI. For all three primers, the first target was Olea europaea. They did not capture O. fragrans! I know that Olea europaea and O. fragrans belong to the same family (Oleaceae). Could you please explain why your primers are not specific to your plant?
